# The Impact of Human Milk Oligosaccharides on Antibiotic-Induced Microbial Dysbiosis and Gut Inflammation in Mice

**DOI:** 10.3390/antibiotics14050488

**Published:** 2025-05-10

**Authors:** Kristine Rothaus Christensen, Torben Sølbeck Rasmussen, Caroline M. Junker Mentzel, Sofie Kaas Lanng, Elena Tina Gabriella Meloni, Hanne Christine Bertram, Camilla Hartmann Friis Hansen, Axel Kornerup Hansen

**Affiliations:** 1Department of Veterinary and Animal Sciences, University of Copenhagen, Ridebanevej 9, DK-1870 Frederiksberg C, Denmark; kristine.christensen@dsm-firmenich.com (K.R.C.); caroline@sund.ku.dk (C.M.J.M.); camfriis@sund.ku.dk (C.H.F.H.); 2Dsm-Firmenich, Kogle Allé 4, DK-2970 Hørsholm, Denmark; 3Department of Food Science, University of Copenhagen, Rolighedsvej 26, DK-1958 Frederiksberg C, Denmark; torben@food.ku.dk (T.S.R.); elenatinagabriella.meloni@biologo.onb.it (E.T.G.M.); 4Department of Food Science, Aarhus University, Agro Food Park 48, DK-8200 Aarhus N, Denmark; skl@clin.au.dk (S.K.L.); hannec.bertram@food.au.dk (H.C.B.); 5Department of Biotechnology and Biosciences BtBs, Piazza dell’Ateneo Nuovo, I-1-20126 Milan, Italy

**Keywords:** oligosaccharides, ampicillin, microbiota, mice, *Bacteroides*, *Lactobacillus*

## Abstract

**Background/Objectives**: Antibiotics have a significant impact on the gut microbiota, and we hypothesized that human milk oligosaccharides may alleviate antibiotic-induced gut microbiota dysbiosis. **Methods**: Six groups of eight mice were administered drinking water with or without ampicillin for one week. We then introduced the human milk oligosaccharide 2′-fucosyllactose (2′FL), either alone or in combination with difucosyl-lactose (DFL), for two weeks after the termination of ampicillin treatment. **Results**: Ampicillin reduced microbiota diversity and the abundance of specific bacteria. One week after the termination of ampicillin treatment, the 2′FL + DFL mixture counteracted the ampicillin-induced reduction in diversity, although this effect was not sustained. Over the subsequent two weeks, the 2′FL + DFL mixture had a significant impact on the relative abundances of *Lactobacillus* spp. and *Bacteroides* spp. Ampicillin also reduced caecal propionate levels, downregulated the gene *Gzmb* for Granzyme B, and upregulated the gene *Reg3a* for Regenerating islet-derived protein 3 alpha, all of which were counteracted by the 2′FL + DFL mixture. Ampicillin had a minor impact on ileal cytokine levels. The 2′FL + DFL mixture showed a cytokine effect indicating reduced adaptive and innate inflammation. Ampicillin reduced water intake and growth in the mice. The oligosaccharides did not affect water intake, but the 2′FL + DFL mixture slightly reduced body weight. **Conclusions**: The 2′FL + DFL mixture appears to hold potential for counteracting some of the side effects of ampicillin treatment.

## 1. Introduction

Although long-term treatment with a broad-spectrum antibiotic such as ampicillin is considered safe [1], antibiotics exert several other effects in the host beyond merely reducing the growth of target microbes.

Especially when administered orally, antibiotics may inhibit specific members of the gut microbiota. Over time, antibiotics may promote the overgrowth of certain bacteria, such as Proteobacteria, leading to dysbiosis [2,3]. In a worst-case scenario, antibiotics may trigger *Clostridioides difficile*-associated diarrhea [4]. Ampicillin appears to be the most effective component of antibiotic cocktails used to induce a pseudo-germ-free state in mice, although it often fails to eradicate some microaerophilic Gram-positive bacteria [5]. Moreover, some bacteria are only suppressed by ampicillin and may proliferate following the cessation of treatment [6].

In the human colonic microbiota, several metabolic functions—such as fermentation products and the synthesis of enzymes—are influenced by ampicillin treatment [7]. Similarly, in the caecal contents of mice, treatment with amoxicillin (the more absorbable formulation of ampicillin) reduces concentrations of branched short-chain fatty acids (SCFA), while increasing the levels of metabolites, such as formate, lactate, and various amino acids and their derivatives [3].

For years, probiotics have been considered a means of correcting antibiotic-induced dysbiosis. In the fecal and cecal contents of amoxicillin-treated mice, supplementation with *Bifidobacterium animalis* subsp. *lactis* has been associated with increased abundances of *Streptococcaceae* and reduced abundance of *Erwiniaceae* [3]. In humans, administration of *Lacticaseibacillus rhamnosus* GG and *Saccharomyces cerevisiae boulardii* appears to elevate SCFAs, such as acetate, propionate, butyrate, and lactate [8].

In mammals, up to 10^13^ bacteria may have colonized the gut [9], supported by oral tolerance maintained by the gut immune system [10]. It may, therefore, be optimistic to expect a few beneficial bacteria to colonize in such a densely populated microbiotic environment. Although engineered probiotics may hold promise as future therapeutics, their application still faces several challenges [11]. Feeding prebiotics may, therefore, be a more rational approach. Prebiotics are chemically defined, non-digestible, and non-nutritive food components that promote the growth or activity of beneficial microbes [12]. Some of the most common examples are oligosaccharides—short-chain carbohydrates—which can be sourced either from plants [13] or human breast milk [14]. Plant-based oligosaccharides, such as mannan-, xylose-, and galacto-oligosaccharides, have been shown to support gut microbiota recovery following antibiotic disturbance by providing nutritional support for *Bifidobacterium* spp., *Lactobacillus* spp., *Akkermansia* spp., and *Bacteroides* spp., and such prebiotic treatment reduces intestinal inflammation and damage in ampicillin-treated mice [8,15,16].

Human milk oligosaccharides (HMOs), at concentrations of 10–15 g per liter (g/L), are the third most abundant component of human breast milk [17]. The over 200 identified structures of HMOs in breast milk fall into three main categories: fucosylated, sialylated, or neutral-core types [18]. The fucosylated HMO 2′-fucosyllactose (2′FL) is the most abundant, with difucosyl-lactose (DFL) also present in significant amounts [14,17,18]. HMOs serve as an energy source for certain highly specific beneficial gut bacteria and inhibit the growth of specific pathogens by mimicking sugar-structure receptors [19]. They are metabolized into SCFAs [20], which serve as an energy source for enterocytes and lower colonic pH—conditions that favor the growth of beneficial and symbiotic bacteria [20]. In addition, HMOs enhance the gut mucosal barrier [21], block viral infections through receptor mimicry [22], and modulate the gene expression in intestinal cells [23]. This might have the potential to alleviate human diseases such as allergic airway inflammation [24] and pneumococcal pneumonia [25]. Consequently, there is growing scientific interest in elucidating prebiotic mechanisms, and the food industry increasingly views them as potential components of functional foods [26]. Over the past decade, large-scale production of HMOs via microbial synthesis has become feasible [27]. However, little is known about how HMOs affect ampicillin-induced gut microbiota dysbiosis and its related consequences.

Mice are a convenient and cost-effective tool for preclinical studies under well-controlled conditions prior to more complex and costly trials in human patients, although in certain research areas their validity may be limited [28]. In addition to the advantage of having enough units to allow for proper statistical analysis, another benefit of using mice is the wealth of literature demonstrating the efficacy of oligosaccharides in mice [24,29,30,31,32,33,34]. However, a major disadvantage is that mouse milk primarily contains sialylated oligosaccharides, whereas human milk primarily contains fucosylated oligosaccharides, which preferentially support different bacterial taxa [32].

In this study, we treated mice with ampicillin and additionally administered 2′FL, either alone or in combination with DFL. We hypothesized that these HMOs might alleviate gut microbiota dysbiosis. Our primary readout was microbiota characterization via 16S amplicon sequencing. Secondary readouts included assessment of intestinal cytokines, gene expression, and SCFA concentrations. The mixture of 2′FL and DFL appeared to mitigate some of the adverse effects of ampicillin treatment.

## 2. Results

### 2.1. Ampicillin Reduced the Microbiota Diversity

We administered drinking water with or without ampicillin for a week to six groups of eight mice. We mixed in the HMO 2′-fucosyllactose (2′FL) alone or in combination with difucosyl-lactose (DFL) for two weeks after cessation of ampicillin treatment. The control group was given clean water throughout the entire duration of this study (Figure 1).

The alpha diversity (Shannon index) reflects the number of different bacterial species harbored by each mouse. On days 7, 14, and 21, alpha diversity in ampicillin-treated mice—whether or not they received 2′FL or 2′FL + DFL—was significantly lower than in untreated control mice (Figure 2A). However, on day 7 (the day of ampicillin cessation), alpha diversity was significantly higher in the 2′FL + DFL-treated group compared to ampicillin-only mice. In contrast, on day 14 (one week post-treatment), the alpha diversity in the 2′FL + DFL group was significantly lower than that in ampicillin-only mice (Figure 2A). By day 21 (two weeks after ampicillin cessation), only mice treated with 2′FL—not those treated with 2′FL + DFL—exhibited significantly lower alpha diversity compared to the ampicillin-only group (Figure 2A). At no time point did HMOs influence alpha diversity in mice not treated with ampicillin (Figure 2A). Bray–Curtis indices (beta diversity) reflect differences in microbiota composition between individual mice. The closer the clustering, the more similar the microbiota compositions. Ampicillin-treated mice clustered distinctly from untreated controls on days 7, 14, and 21 (Figure 2B). Among ampicillin-treated mice, there was no significant clustering based on HMO treatment. However, among mice not treated with ampicillin, clustering differed significantly between HMO-treated and non-treated groups (Figure 2B).

### 2.2. Ampicillin Significantly Reduced the Abundance of a Range of Bacteria, Which for Some Specific Bacteria Was Counteracted by HMOs

Ampicillin significantly reduced the relative abundances of numerous bacterial families. The 2′FL + DFL mixture significantly increased the relative abundances of specific bacterial families, whereas the effect of 2′FL alone was more limited (Table 1 and Appendix A). Nevertheless, its influence remained detectable—particularly in mice treated solely with ampicillin—even one (Figure 3), two (Figure 4), and three weeks (Figure 5) after cessation of treatment. On day 14, i.e., two weeks after the termination of ampicillin administration, the reductions in Lactobacillaceae and Coriobacteriaceae induced by ampicillin appeared to be counteracted by the 2′FL + DFL mixture (Figure 4A,B). At the genus level, this effect was primarily driven by increases in Lactobacillus spp. (Figure 6). Furthermore, *Bacteroides* spp. were promoted in ampicillin-treated mice receiving the 2′FL + DFL mixture (Figure 6). On day 14, both HMOs appeared to increase the abundance of Enterobacteriaceae in ampicillin-treated mice (Table 1 and Figure 4C), an effect driven at the genus level by *Escherichia* spp. (Figure 6).

### 2.3. The 2′FL + DFL Mixture Reduced Adaptive and Innate Inflammation

Following treatment cessation, we assessed ileal cytokine levels to investigate whether ampicillin or the HMOs—either alone or in combination—had an impact on gut inflammation. The false discovery rate (FDR)-corrected effect of ampicillin was only borderline significant and was limited to a few innate immune-related cytokines and chemokines: interleukin-1β (IL-1β), macrophage inflammatory protein-1α (MIP-1α), and interferon-γ (IFN-γ) (Figure 7A,B). The combination of 2′FL and DFL significantly reduced IL-1β (q = 0.005; Figure 7A–C) and MIP-1α (q = 0.005; Figure 7A,B,D) and showed a borderline reduction in IFN-γ (q = 0.089; Figure 7A,B,E), although no significant post hoc differences were detected. 2′FL alone significantly reduced IFN-γ (q = 0.033; Figure 7A,B), but again without any post hoc significance (Figure 7E). Notably, ampicillin appeared to attenuate the effects of 2′FL on IL-1β (*p* = 0.023; Figure 7C) and MIP-1α (*p* = 0.0003; Figure 7D).

### 2.4. Cecal Propionate Levels Were Reduced by Ampicillin and Increased by the 2′FL/DFL Mixture

Cecal propionate levels were significantly reduced by ampicillin treatment (q = 0.000, Figure 8) but were increased by the 2′FL + DFL mixture (q = 0.042; Figure 8). In comparison to mice receiving the 2′FL + DFL mixture, propionate levels were significantly lower in ampicillin-treated mice (*p* = 0.002) and borderline lower in ampicillin-treated mice supplemented with 2′FL alone (*p* = 0.053). However, this reduction was not observed in ampicillin-treated mice receiving the 2′FL + DFL mixture (Figure 8C). No significant effects were observed on cecal acetate or butyrate levels.

### 2.5. Ampicillin Downregulated Ileal Gzmb and Upregulated Ileal Reg3a

A panel of ileal genes was analyzed to determine whether ampicillin affected gene expression and whether the HMOs modulated this effect (Appendix A). Prior to FDR correction, ampicillin significantly altered the expression of most genes (Figure 9A). However, following FDR correction, only two effects remained significant: a downregulation of *Gzmb* (*p* = 0.004, Figure 9A,B) and an upregulation of *Reg3a* (*p* = 0.004, Figure 9A,C).

### 2.6. Ampicillin Reduced the Growth of the Mice

To assess whether ampicillin or HMOs affected growth or water intake, the mice and their water bottles were weighed before and after use. Ampicillin had a significant effect on body weight across all three time periods (Appendix A; *p* = 0.000, *p* = 0.001, *p* = 0.001). The 2′-FL + DFL mixture showed a borderline reduction in body weight (Figure 2; *p* = 0.068, *p* = 0.068, *p* = 0.089), whereas 2′-FL alone did not appear to influence growth. Ampicillin also significantly reduced water consumption (Appendix A; *p* = 0.001), while water intake was unaffected by HMO supplementation (Appendix A).

## 3. Discussion

We hypothesized that HMOs might alleviate ampicillin-induced gut microbiota dysbiosis in mice. As expected, the composition and diversity of the gut microbiota, the various specific bacterial abundances, caecal levels of SCFAs, and gut gene expressions were strongly affected by ampicillin. Supplementing ampicillin-treated mice with the 2′FL + DFL mixture mitigated the effect on specific bacteria, such as *Bacteroides* and *Lactobacillus* spp., and in the early phase immediately after cessation of ampicillin treatment, it also mitigated the ampicillin-induced reduction in gut microbiota diversity. It is difficult to explain the mechanism behind the enhanced effect of the combination of the two HMOs compared to 2′FL alone. In accordance with our study, the combination of two HMOs has previously been shown to be the most efficient for eliciting effects through the gut microbiota [35]. It was surprising that the 2′FL + DFL mixture mitigated ampicillin effects on *Lactobacillus* spp., as such an effect is not usually expected from 2′FL and DFL [36]. Various *B. longum* subspecies as well as other *Bifidobacterium* spp. are known to be able to utilize 2′FL and DFL [37]. In this study, *Bifidobacterium* spp. were detected in the gastrointestinal tract of the mice, but the HMOs did not have any significant effect on them. The effect of HMOs is known to be very specific, down to the genus level. Our sequencing only identified to the level of *Bifidobacterium* spp., but mice typically harbor *B. pseudolongum subsp. animalis* [38], which are unable to utilize HMOs [39]. Our data, therefore, indicate that although mice are naturally colonized with *Bifidobacterium* spp., they are not necessarily always useful as models for studying the very specific impact of HMOs, as HMO utilization is very specific down to the strain level of bacterial species. Furthermore, mouse milk is dominated by a high level of sialylated oligosaccharides, such as 3′-sialyllactose (3′SL) and 6′ sialyllactose (6′SL) [32]. In contrast, human milk is dominated by fucosylated oligosaccharides, such as 2′FL [32]. Therefore, the anti-inflammatory effect of the HMOs must have been a result of an effect on other anti-inflammatory bacteria, such as *Lactobacillus* spp. and *Bacteroides* spp. [40]. *Bacteroides fragilis*, *B. vulgatus*, and *B. thetaiotaomicron* all utilize both 2′FL and DFL [36,41,42]. The ability of *Lactobacillus* spp. to metabolize HMOs has mostly been described for species that have now been reclassified as *Lacticaseibacillus* spp. [43], most of which do not utilize 2′FL and DFL effectively [36,44]. *Lactobacillus* spp. in the mice of this study were not identified to species or strain level. There are strains of *L. acidophilus*, such as FUA3191, that utilize HMOs [45]. It might be speculated that mouse studies would be more translational if mice were supplied with human bacterial strains on which an impact is expected. However, this remains challenging. *Bifidobacterium* spp., known to be probiotic in humans, do not necessarily colonize very well in SPF mice [46], and therefore, must be constantly fed with no guarantee that they will affect the host [46]. On the other hand, studying oligosaccharide effects in animal models with no *Bifidobacterium* spp. can be advantageous, as more discrete impacts on other bacterial species can be elucidated, such as the impact on *Prevotella* spp. in *Bifidobacterium*-free mice [31] and the impact on *Akkermansia* spp. in horses [47].

Propionate is a product of carbohydrate fermentation in the gut and is a substrate for hepatic gluconeogenesis [48]. *Bacteroides* spp. are well-known producers of propionate [49,50], whereas *L. acidophilus* primarily converts HMOs into lactate and butyrate, producing only trace amounts of propionate [51].

The levels of IFN-γ, which is produced by the ileal T cells [52], were significantly reduced by 2′FL. The 2′FL + DFL mixture significantly induced a significant reduction in the levels of the cytokine IL-1β and the chemokine MIP-1α. IL-1β is a pro-inflammatory cytokine produced by activated macrophages, monocytes, and a subset of dendritic cells [53], while MIP-1α is produced by macrophages and monocytes following stimulation by lipopolysaccharides from the cell walls of Gram-negative bacteria through the Toll-like receptor 4 (TLR4) or pro-inflammatory cytokines such as IL-1β [54]. Certain *Bacteroides* spp. are known to facilitate the production of the secondary bile acid, hyodeoxycholic acid, which is anti-inflammatory, as it reduces TLR4 stimulation [55,56]. Administration of hyodeoxycholic acid to mice reduces IL-1β levels [56]. HMO effects in this study are, therefore, most likely mediated via *Bacteroides*.

2′FL increased the abundance of *Escherichia* spp. on day 14, i.e., one week after ampicillin cessation, although 2′FL has previously been shown to reduce the abundance of *E. coli* [57]. The anti-inflammatory profile of mice administered the 2′FL + DFL mixture does not indicate an inflammatory effect of *Enterobacteriaceae*, such as *Escherichia* spp. Microbiota characterization in live mice must be conducted using fecal samples. By mimicking host sugar receptors [19], HMOs probably encourage the accumulation of Enterobacteriaceae in the feces during the early stages of treatment.

Ampicillin had a significant effect on gene expressions, whereas HMOs did not. Ampicillin downregulated *Gzmb* and upregulated *Reg3a*. *Gzmb* encodes a member of the granzyme subfamily of proteins, Granzyme B. This is secreted by natural killer cells and cytotoxic T lymphocytes [58]. In addition to targeting cell apoptosis [59], Granzyme B also processes cytokines and degrades extracellular matrix proteins related to, for example, skin inflammation [60] and wound healing [58]. However, ampicillin only had a borderline effect on cytokine production. *Reg3a* encodes a bactericidal C-type lectin, regenerating islet-derived protein 3 alpha, which is primarily expressed in the gut and related tissues (highest in the pancreas) [61]. It has a bactericidal effect when it binds to peptidoglycan in the cell walls of Gram-positive bacteria [62].

It is a limitation to this study that not all members of the microbiota in commercial laboratory mice represent human-relevant bacterial species and subspecies, as may be the case for *Bifidobacterium* spp. and *Lactobacillus* spp. Also, as in many mouse studies, not all bacteria in this study were identified at the species level. The dose of ampicillin normally used for mice is higher than the most common human dose when calculated per kilogram of body weight, and therefore, the effects of ampicillin may appear more pronounced in mice than would be the case in humans. Overall, this limits the predictive validity of this study, as it remains unclear whether the bacterial taxa affected in mice would also be similarly affected in humans. Finally, it is a limitation that only female mice were used. The rationale for this was to prevent cage fighting [63], but this unfortunately limits the validity of results for both sexes.

## 4. Materials and Methods

### 4.1. Animals

In this study, 48 female BALB/cJBomTac mice (Taconic Biosciences, Ll. Skensved, Denmark) were used. The mice were divided into six groups, with four mice housed per cage. Three-week-old mice were earmarked and randomly assigned by the animal technicians into six groups of eight mice, housed with four mice housed per cage (open-type 1284L Eurostandard II L; Techniplast, Varese, Italy) at the University of Copenhagen’s AAALAC-accredited barrier-protected experimental rodent facility. The number of animals was based on the effect and variation in previous studies, in which eight mice had been shown to be the minimum needed to have enough power to show statistical significance when performing the individual analyses. A single animal in the ampicillin + 2′FL + DFL group was found dead in the cage without a diagnosis after necropsy on day 15. No further animals were excluded during this study. The facility maintained a controlled and consistent environment, with a temperature of 22 °C ± 2 °C, a relative humidity of 45–55%, and a 12/12 light–dark cycle with lights on at 0600–1800 h. Each cage contained aspen bedding material (Tapvei, Harjumaa, Estonia), a handling tube (Datesand, Stockport, UK), cotton nestlets (Ancare, Bellmore, NY, USA), an aspen, mouse-sized chew block (Tapvei, Harjumaa, Estonia), a play tunnel, and a cardboard house (Brogaarden, Lynge, Denmark). Cages were cleaned and provided with new bedding and nest material weekly. Enrichment items were only replaced if they had been shredded by the mice. The mice had unrestricted access to diet (Altromin 1324, Altromin, Lage, Germany) and bottled tap water during this study.

### 4.2. Study Setup

The mice acclimatized in the unit for 1 week. Hereafter the control group received clean water, while the other five groups were administered the test substances, 2′FL, 2′FL + DFL, and/or ampicillin in their drinking water (Figure 1). Bottles were changed biweekly, and water intake per cage was measured (Figure 1). Ampicillin (Sigma-Aldrich A9518-25G, Søborg, Denmark) was administered as 1 g/L drinking water, and only during the first study week (Figure 1). This ampicillin dose is known to be efficient, as it is used in most studies on microbiota eradication in mice [64]. Compared to a human dose, it is on the high end if recalculated for mice according to metabolic weight [65]. HMOs (dsm-firmenich, Hørsholm, Denmark) were dissolved as 9.88 g/L 2′FL (Technical grade, Batch No.: CPN2931300215) or 12.36 g/L 2′FL + DFL (Technical grade, Batch No.: CPN6317 1,000,217 FDT) and given for all three weeks to the relevant groups (Figure 1). The HMO dose was calculated from the assumption that each mouse drinks approximately 3.5 mL of water per day and weighs approximately 20 g during this study. Mature human milk contains 10–15 g HMO per liter [66], and the mean daily intake for an infant is 670 mL [67], which gives an average intake of approximately 8–10 g per day. Recalculation of a mouse dose from a human dose of 10 g per day according to metabolic weight [65] gives a dose of 140 mg/kg body weight. Our HMO dose is equivalent to anti-inflammatory doses in other studies [33]. However, it is not uncommon in mouse studies to use doses much higher than what is human equivalent [34,68]. At baseline and after each of the three study weeks, each individual mouse was weighed, and feces were individually sampled. After three weeks all mice were anesthetized using 0.1 mL/10 g body weight of a mixture of 25% Midazolam (5 mg/mL midazolam, B. Braun, Melsungen, Germany) and 25% Hypnorm (0.315 mg/mL of fentanyl citrate and 10 mg/mL of fluanisone, Skanderborg Apotek, Skanderborg, Denmark). Blood was sampled from the retroorbital plexus of the eye into a heparin-coated capillary (Kruuse, Langeskov, Denmark). Hereafter, the mice were euthanized by cervical dislocation. Approximately 0.5 cm of ileums (most distal) and colons (most proximal) were sampled. These were rinsed with PBS, placed in RNAlater (Fisher Scientific Biotech Line 10427114, Slangerup, Denmark) on dry ice, and stored at −80 °C for quantitative polymerase chain reaction (qPCR) analysis. Approximately 0.5 cm of ileum tissue (middle part) was sampled, rinsed with PBS, frozen on dry ice, and then stored at −80 °C prior to cytokine analysis.

### 4.3. Microbiota Characterization

Feces were stored at −80 °C until 16S rRNA gene amplicon sequencing. The Bead-Beat Micro AX Gravity kit (A&A Biotechnology, Cat. No. 106–100 mod.1) was used to extract bacterial DNA from the fecal pellet as well as from included positive (bacterial mock community) and negative (MiliQ water) controls by following the instructions of the manufacturer. The final purified DNA was stored at −80 °C, and the DNA concentration was determined using the Qubit HS Assay Kit (Invitrogen, Waltham, MA, USA) on the Qubit 3 Fluorometric Quantification device (Invitrogen). The primers (NXt_388_F: 5′-TCGTCGGCAG CGTCAGATGT GTATAAGAGA CAGACWCCTA CGGGWGGCAG CAG-3′ and NXt_518_R: 5′-GTCTCGTGGGC TCGGAGATGTG TATAAGAGAC AGATTACCGC GGCTGCTGG-3, Integrated DNA Technologies; Leuven, Belgium) compatible with the Nextera Index Kit were used to amplify the V3 region of the 16S rRNA gene. The first PCR reaction: mixing 12 μL of AccuPrime™ SuperMix II (Life Technologies, Carlsbad, CA, USA), 0.5 μL of primer mix (10 μM), and 5 μL of genomic DNA (~20 ng/μL) to a total volume of 20 μL. Cycling was run on a SureCycler 8800 under the following conditions: 95 °C for 2 min; 33 cycles of 95 °C for 15 s, 55 °C for 15 s, and 68 °C for 30 s; followed by the final step at 68 °C for 5 min. Furthermore, 12 μL of Phusion High-Fidelity PCR Master Mix (Thermo Fisher Scientific, Waltham, MA, USA), 2 μL of corresponding P5 and P7 primers (Nextera Index Kit, San Diego, CA, USA), 2 μL of PCR product, and nuclease-free water for a total volume of 25 μL were mixed. The 2nd cycling conditions were 98 °C for 1 min; 12 cycles of 98 °C for 10 s, 55 °C for 20 s, and 72 °C for 20 s; and 72 °C for 5 min. AMPure XP beads (Beckman Coulter Genomic, Copenhagen, Denmark) were used to purify the PCR products. AMPure XP beads were mixed with the PCR product and incubated at room temperature for 5 min and mounted on a magnetic rack for 2 min. The supernatant was discarded. The beads were now washed with 150 µL of 80% ethanol twice without disturbing the beads. The samples were removed from the magnetic rack, and the beads were mixed with 25 µL of PCR-grade water and incubated at room temperature for 2 min. The PCR tube was then mounted to the magnetic rack again for 2 min before the sampling of clean DNA products. The bacterial community composition was determined by Illumina NextSeq-based high-throughput sequencing of the 16S rRNA gene V3 region. Quality control of reads, dereplicating, purging of chimeric reads, and constructing zOTUs were conducted with the UNOISE pipeline [69] and taxonomically assigned with Sintax [70]. Taxonomical assignments were obtained using the EZtaxon for the 16S rRNA gene database [71]. Code describing this pipeline can be accessed at github.com/jcame/Fastq_2_zOTUtable. The average sequencing depth after quality control at zOTU table assembly (Accession: PRJEB80473, available at ENA) for the fecal 16S rRNA gene amplicons was 45,240 reads (min. 9175 reads and max. 93,913 reads). Prior to analysis, the zOTU dataset was purged of zOTUs that were detected in less than 5% of the samples, but the resulting dataset still maintained 99.5% of the total reads. Cumulative sum scaling (CSS) [72] was applied for the analysis of beta diversity to counteract that the majority of count values were only represented by a few zOTUs. CSS normalization was performed using the R software using the metagenomeSeq package [72]. Alpha diversity analysis was based on raw read counts, and its statistics were based on ANOVA, while beta diversity was represented by Bray–Curtis dissimilarity, and its statistics were based on PERMANOVA. Both were FDR corrected. R version 4.4.2 was used for subsequent analysis and presentation of 16S rRNA gene amplicon data. The main packages used were phyloseq [73], vegan [74], deseq2 [75], ampvis2 [76], ggpubr [77], mctoolsr (https://leffj.github.io/mctoolsr/) (accessed on 6 May 2025), and ggplot2 [78].

### 4.4. Cytokines

After sampling, blood was transferred to a sterile Eppendorf tube (Hounisen, Skanderborg, Denmark) and placed at room temperature for a maximum of 20–30 min before transfer to 5 °C. The samples were centrifuged at 8000× *g* for 10 min, and the supernatant (plasma/serum mix) was collected, frozen on dry ice, and stored at −80 °C. The ileum tissue was weighed prior to homogenization and then homogenized three times with 5 min intervals (on ice) in a buffer containing EDTA and phosphatase and proteinase inhibitors (MSD TRIS lysis buffer (R60TX-3) and MSD inhibitor pack (R70AA-1); Meso Scale Diagnostics, Rockville, US; 10 µL per 1 µg of tissue) by bead-beating for 45 s at 6 M/S (Bead-Beat Micro AX Gravity kit (A&A Biotechnology 106-100, Gdynia, Poland). Samples were incubated at 4 °C for 20–30 min before centrifugation at 7500× *g* for 5 min, and the supernatant was transferred to Eppendorf tubes and stored at −80 °C. After thawing, Meso Scale cytokine analysis was conducted with the mouse pro-inflammatory cytokine panel (K15048D) and the cytokine panel I (K15245D) (Meso Scale, Rockville, MD, USA).

### 4.5. Proton (^1^H) NMR Spectroscopic Analysis of Short-Chain Fatty Acids (SCFAs)

^1^H NMR spectroscopy was conducted using a Bruker Avance III 600 MHz spectrometer operating at a ^1^H frequency of 600.13 MHz with a 5 mm ^1^H TXI probe (Bruker BioSpin, Rheinstetten, Germany). Samples of 20 mg freeze-dried caecal content were solubilized in demineralized H_2_O, thoroughly whirl mixed, and centrifuged for 30 min at 20,000× *g*. A volume of 400 µL was added to 100 µL of phosphate buffer (0.75 M, pH = 7.4) and 100 µL of D_2_O containing 0.025% trimethylsilyl-propanoic acid (TSP). ^1^H-NMR spectra were obtained at 298 K using a one-dimensional (1D) nuclear overhauser enhancement spectroscopy (NOESY) presat pulse sequence (noesypr1d) to ensure water suppression. The acquisition parameters used were 128 scans, spectral width 7289 Hz (12.15 ppm), acquisition time 2.25 s, 32K data points relaxation delay 5 s. Prior to Fourier transformation, the free induction decays (FIDs) were multiplied by a line-broadening function of 0.3 Hz. The spectra obtained were subjected to baseline correction and phase correction in TopSpin 3.0 (Bruker BioSpin, Billerica, MA, USA). Chenomx NMR Suite 8.13 (Chenomx Inc., Edmonton, AB, Canada) was used for metabolite assignment and quantification of acetate, propionate, and butyrate using TSP as an internal quantification standard.

### 4.6. Fluidigm Biomark High-Throughput Gene Expression Analysis

In this study, 0.5 cm ileum pieces were transferred to tubes (FastPrep^®^ 50-76-200, Mpbio, Navi Mumbai, Maharashtra) containing 0.6 g of glass beads (G4649, Sigma-Aldrich), 600 µL of lysis binding solution concentrate (AM1830, Invitrogen™), and 0.7% beta-mercaptoethanol (M6250, Sigma-Aldrich). The samples were homogenized using the FastPrep-24™ Classic Instrument (Mpbio) with 4× (45 s at speed 6.5) runs. The homogenate was centrifuged at 16,000× *g*, and the supernatant was frozen at −20 °C overnight. Furthermore, 100 µL of homogenate was purified to RNA using the MagMax™ Express Magnetic Particle Processor (Applied Biosystems ™, Waltham, MA, USA) using the manufacturer’s instructions (kit and protocol: AM1830, Invitrogen™). RNA purity and concentration were assessed using NanoDrop ND-1000 (NanoDrop Technologies, Waltham, MA, USA), and RNA integrity was assessed by 1.4% agarose gel to check for clear 18S RNA and 28S RNA bands on the gel without smearing. cDNA synthesis was performed in duplicate from 500 ng total RNA using the Quantitect Reverse Transcription kit (Qiagen, Hilde, Germany). Negative controls were made by excluding the reverse transcriptase in the cDNA synthesis reaction from one sample (-RT control). cDNA was diluted in a ratio of 1:8 and stored at −80 °C before qPCR. Genes involved in inflammation, gut microbiota signaling, and gut barrier from a dynamic in-house gut–immunity panel previously published were used [79] (Appendix A). The primers (Sigma-Aldrich) were designed to span an intron if possible and yield products around 75–200 nucleotides long using Primer3 (http://bioinfo.ut.ee/primer3/, (accessed on 6 May 2025)) or Primer Blast (https://www.ncbi.nlm.nih.gov/tools/primer-blast/, (accessed on 6 May 2025)) with standard settings [80,81]. qPCR was performed using the Biomark HD system (Fluidigm Corporation/Standard Bio Tools, San Francisco, CA, USA) on 2x 96.96 IFC chips on pre-amplified cDNA duplicates using the manufacturer’s instructions with minor adjustments as previously described [82]. Melting curves were assessed prior to further analysis, and primer assays yielding multiple products and/or assays yielding a product in the –RT samples were abolished. A calibration curve made from a 5× dilution row of a pool of the pre-amplified cDNA was used to calculate primer efficiency. Primer assays with efficiencies between 80 and 110% were accepted for further analysis. All qPCR data processing prior to statistics was performed in Genex6 (multiD Analysis AB, Västra Frölunda, Sweden). Furthermore, 66 candidate genes and 10 reference genes were assessed. The reference genes were analyzed for stable expression using the geNorm and NormFinder algorithms [83,84]. *Hpr*, *Pgk1*, *Ppia*, *Sdha*, *Tuba*, and *Ywhaz* were most stable and used as reference genes. Candidate gene expression in cycle of quantification values (Cq) was normalized to the reference genes, and cDNA duplicates were averaged. Relative expression of the lowest expressed sample per tissue was set to 1. The data were log2-transformed before statistical analysis in R using linear models with either control or ampicillin as control groups.

### 4.7. Statistics

Weight and water intake data were evaluated as area under the curve (AUC) for the periods 0–7 days, 0–14 days, and 0–21 days. Data from weight, water intake, cytokines, SCFAs, gene expressions, and relative abundances of bacteria were tested for normality by the Anderson–Darling test and for equality in variation by Levene’s test, and all data were then subsequently tested by a three-way ANOVA for the factors ampicillin, 2′FL (alone), and 2′FL + DFL, followed by Tukey’s post hoc analysis for the weight, cytokines, and SCFAs, while this was not possible for the water intake, as this was measured on the cage level (n = 2) (Minitab v22, Coventry, UK). Moreover, *p*-values for cytokine data were corrected into q-values by false discovery rate (FDR) by the Benjamini/Krieger Yekutieli method [85] (GraphPad Prism v10, Boston, US). For relative bacterial abundances, data were only evaluated by the Kruskal–Wallis test at specific time points if there overall was a significant FDR-corrected q-value for the impact of time, and Dunn’s multiple comparisons were only made if there was a significant q-value at the individual time points (GraphPad Prism, Boston, MA, USA).

## 5. Conclusions

We conclude that ampicillin in mice has a broad reducing impact on the diversity of gut microbiota and the abundances of a range of gut bacteria in mice. A mixture of the HMOs 2′FL and DFL counteracts this reduction in the abundance of *Lactobacillus* spp. and *Bacteroides* spp. This restores cecal propionate levels, which is most likely related to the increased abundances of *Bacteroides* spp. The potential of HMOs to mitigate the adverse effects of ampicillin should be further investigated in human patients to determine whether the efficacy observed in mice is translatable to humans. It would also be relevant with further mouse studies in which some other HMOs, e.g., some sialylated HMOs, are applied.

## Figures and Tables

**Figure 1 antibiotics-14-00488-f001:**
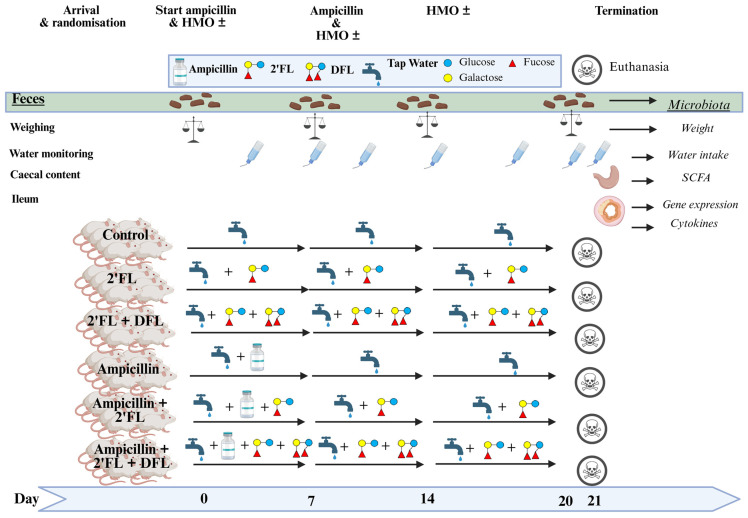
Timeline of a study on 48 female BALB/cJBomTac mice divided into six groups of eight mice, with four mice in each of two cages supplied in their drinking (tap) water with HMOs either as 2′FL alone or 2′FL and DFL in combination, either with or without ampicillin for three weeks from the age of four weeks. The control mice received neither HMO nor ampicillin.

**Figure 2 antibiotics-14-00488-f002:**
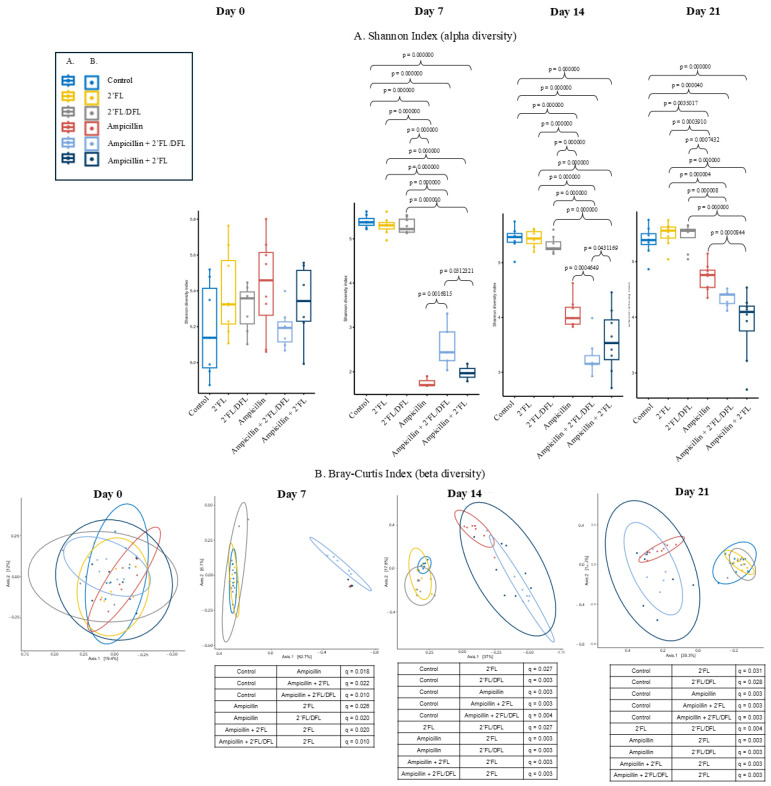
Gut bacteriome diversities of 48 female BALB/cJBomTac mice in six groups of eight mice supplied in their drinking (tap) water with HMOs, either as 2′FL alone or 2′FL and DFL in combination, either with or without ampicillin for three weeks from the age of four weeks at day 0 (before start), day 7 (after ampicillin and HMO treatment), 14 (1 week after ampicillin termination and continuous HMO), and 21 (2 weeks after ampicillin termination and continuous HMO) (**A**) Alpha diversity shown as Bray–Curtis indexes (PERMANOVA with FDR-corrected q-values) (**B**) Beta diversity shown as Shannon indexes (ANOVA with *p*-values of Tukey’s post hoc test).

**Figure 3 antibiotics-14-00488-f003:**
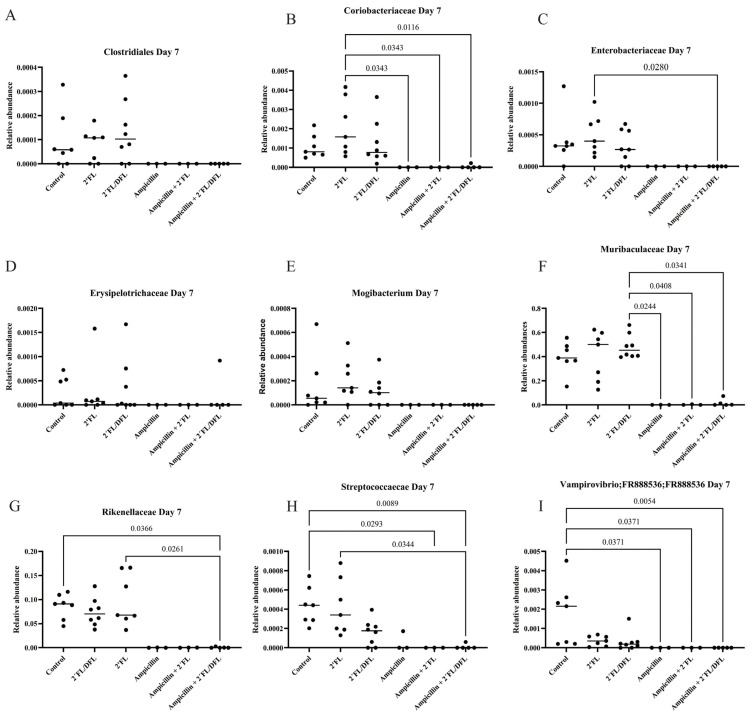
Relative abundances on day 7 of bacterial families Clostridiales (**A**), Coriobacteriaceae (**B**), Enterobacteriaceae (**C**), Erysipelotrichaceae (**D**), Mogibacteirum (**E**), Muribaculaceae (**F**), Rikenellaceae (**G**), Streptococcaceae (**H**) and Vampirovibrio (**I**) in the feces of 48 female BALB/cJBomTac mice were assessed (Figure 1). The mice were divided into six groups of eight, with four mice housed per cage. The cages were supplied with drinking (tap) water containing HMOs—either 2′FL alone or a combination of 2′FL and DFL—administered with or without ampicillin for three weeks starting at four weeks of age. Data were analyzed using a three-way ANOVA, and only those results showing a significant FDR-corrected effect of either ampicillin or one of the two HMO mixtures (q-values) are presented (Table 1).

**Figure 4 antibiotics-14-00488-f004:**
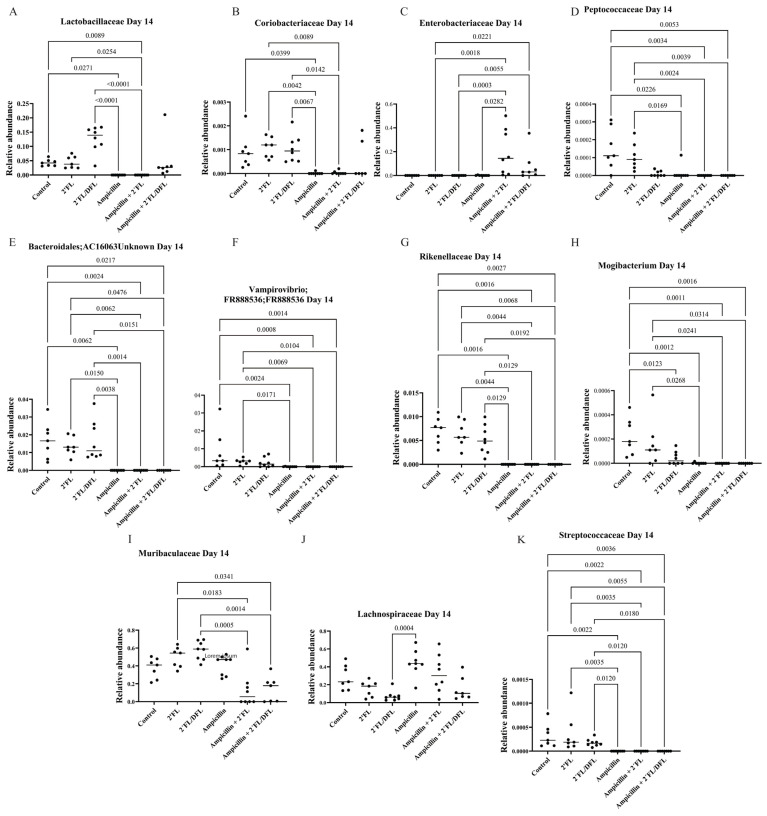
Relative abundances on day 14 of bacterial families Lactobacillaceae (**A**), Coriobacteriaceae (**B**), Enterobacteriaceae (**C**), Peptococcaceae (**D**), Bacteroidales (**E**), Vampirovibrio (**F**), Rikenellaceae (**G**), Mogibacterium (**H**), Muribaculaceae (**I**), Lachnospiraceae (**J**), and Streptococcaceae (**K**) in the feces of 48 female BALB/cJBomTac mice were assessed (Figure 1). The mice were divided into six groups of eight, with four mice housed per cage. The cages were supplied with drinking (tap) water containing HMOs—either 2′FL alone or a combination of 2′FL and DFL—administered with or without ampicillin for three weeks starting at four weeks of age. Data were analyzed using a three-way ANOVA, and only those results showing a significant FDR-corrected effect of either ampicillin or one of the two HMO mixtures (q-values) are presented (Table 1).

**Figure 5 antibiotics-14-00488-f005:**
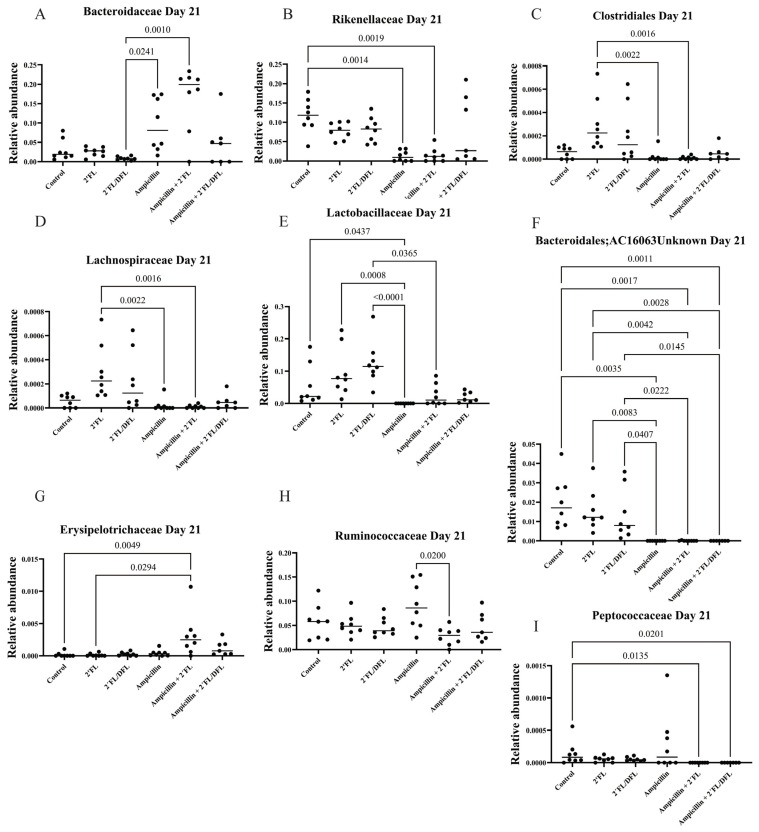
Relative abundances on day 21 of bacterial families Bacteroidaceae (**A**), Rikenellaceae (**B**), Clostridiales (**C**), Lachnospiraceae (**D**), Lactobacillaceae (**E**), Bacteroidales (**F**), Erysipelotrichaceae (**G**), Ruminococcaceae (**H**), and Peptococcaceae (**I**) in the feces of 48 female BALB/cJBomTac mice were assessed (Figure 1). The mice were divided into six groups of eight, with four mice housed per cage. The cages were supplied with drinking (tap) water containing HMOs—either 2′FL alone or a combination of 2′FL and DFL—administered with or without ampicillin for three weeks starting at four weeks of age. Data were analyzed using a three-way ANOVA, and only those results showing a significant FDR-corrected effect of either ampicillin or one of the two HMO mixtures (q-values) are presented (Table 1).

**Figure 6 antibiotics-14-00488-f006:**
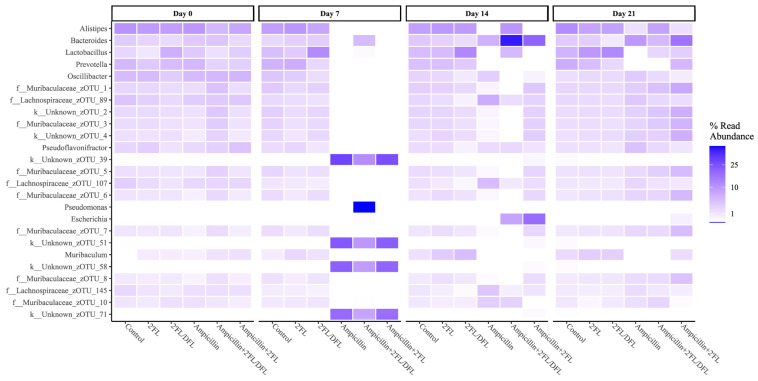
Relative abundances of bacterial genera significantly impacted in the feces of 48 female BALB/cJBomTac mice were assessed (Figure 1). The mice were divided into six groups of eight mice, four mice housed per cage. The cages were supplied with drinking (tap) water containing either 2′FL alone or a combination of 2′FL and DFL administered with or without ampicillin for three weeks starting at four weeks of age.

**Figure 7 antibiotics-14-00488-f007:**
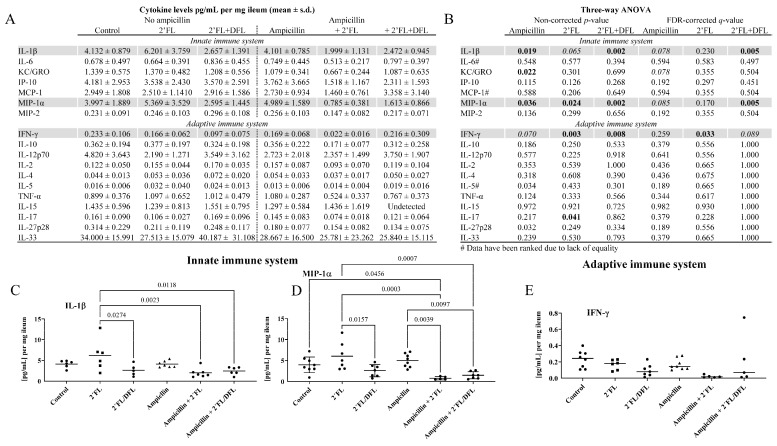
(**A**) Ileal cytokine levels in 48 female BALB/cJBomTac mice were assessed. The mice were divided into six groups of eight mice, four mice housed per cage. The cages were supplied with drinking (tap) water containing either 2′FL alone or a combination of 2′FL and DFL administered with or without ampicillin for three weeks starting at four weeks of age. Data were analyzed using a three-way ANOVA (**B**), and only those results showing a significant FDR-corrected effect of either ampicillin or one of the two HMO mixtures were tested in Tukey’s post hoc test (gray scaled; (**C**–**E**)). Here, *p-* and q-values were considered significant if *p*/q < 0.05 (bold) and borderline if *p*/q < 0.10 (italics). Values on the graphs are *p*-values from Tukey’s post hoc test.

**Figure 8 antibiotics-14-00488-f008:**
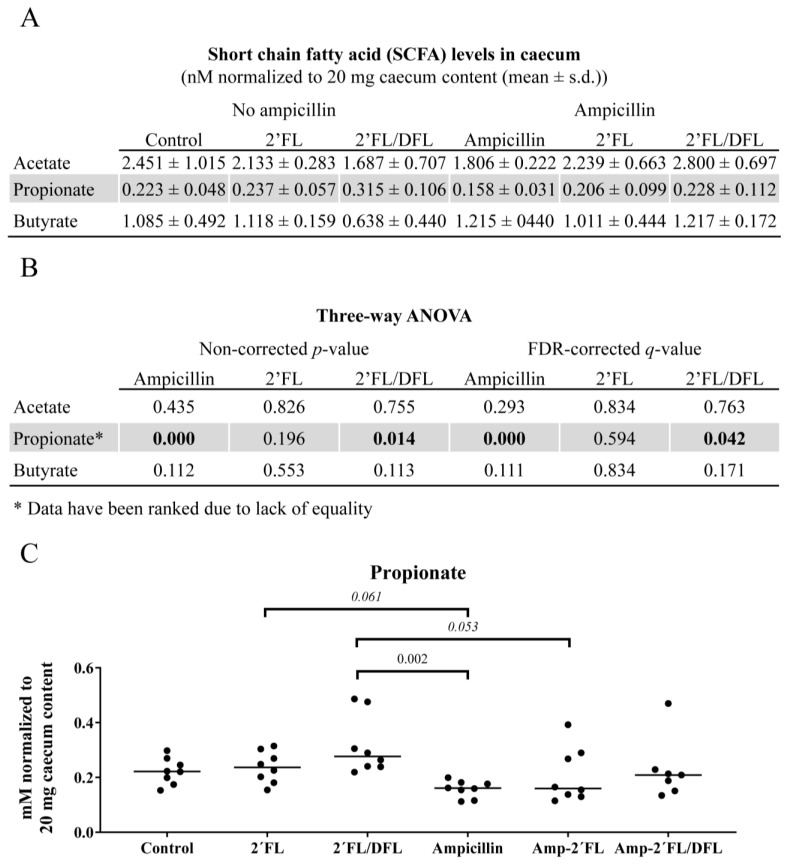
(**A**) Cecal levels of the short-chain fatty acids (SCFAs) acetate, propionate, and butyrate in 48 female BALB/cJBomTac mice were assessed. The mice were divided into six groups of eight mice, four mice housed per cage. The cages were supplied with drinking (tap) water containing either 2′FL alone or a combination of 2′FL and DFL administered with or without ampicillin for three weeks starting at four weeks of age. Data were analyzed using a three-way ANOVA (**B**), and only those results showing a significant FDR-corrected effect of either ampicillin or one of the two HMO mixtures were tested in Tukey’s post hoc test (**C**). Here, *p-* and q-values were considered significant if *p*/q < 0.05 (bold). Values on the graph are *p*-values from Tukey’s post hoc test.

**Figure 9 antibiotics-14-00488-f009:**
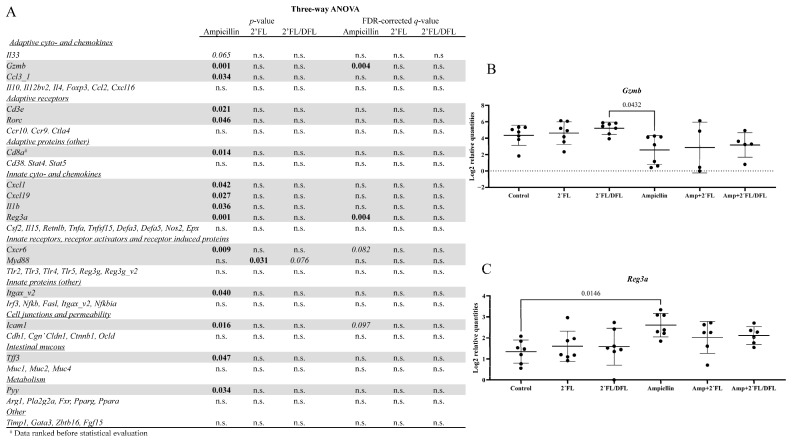
(**A**) Three-way ANOVA on ileal gene expressions in 48 female BALB/cJBomTac mice was assessed. The mice were divided into six groups of eight mice, four mice housed per cage. The cages were supplied with drinking (tap) water containing either 2′FL alone or a combination of 2′FL and DFL administered with or without ampicillin for three weeks starting at four weeks of age. Results are shown as log2 relative quantities in Appendix A. Data were analyzed using a three-way ANOVA (**A**), and only those results showing a significant FDR-corrected effect of either ampicillin or one of the two HMO mixtures were tested in Tukey’s post hoc test (**B**,**C**). Here, *p-* and q-values were considered significant if *p*/q < 0.05 (bold). Values on the graph are *p*-values from Tukey’s post hoc test.

**Table 1 antibiotics-14-00488-t001:** Relative abundances of bacterial families in the feces of 48 female BALB/cJBomTac mice were assessed at specific time points (Figure 1). The mice were divided into six groups of eight, with four mice housed per cage. The cages were supplied with drinking (tap) water containing HMOs—either 2′FL alone or a combination of 2′FL and DFL—administered with or without ampicillin for three weeks starting at four weeks of age. Data were analyzed using a three-way ANOVA, and only those results showing a significant FDR-corrected effect of either ampicillin or one of the two HMO mixtures (q-values) are presented. Data were further evaluated on specific time points only if a significant effect of time was observed (Appendix A). Here, *p-* and q-values were considered significant if *p*/q < 0.05 (bold) and borderline if *p*/q < 0.10 (italics).

	No Ampicillin			Ampicillin			q-Values		
**Day 7**	Control	2′FL	2′FL/DFL	Ampicillin	+.2′FL	+.2′FL/DFL	Ampicillin	2′FL	2′FL/DFL
Actinobacteria; Coriobacteriia; Coriobacteriales; Coriobacteriaceae	0.0010774	0.002086	0.001267	0	0	0.0000435	**0.032**	0.952	0.603
Bacteroidetes; Bacteroidia; Bacteroidales; Muribaculaceae	0.395128	0.407864	0.483105	0.0003331	0.0021549	0.0178059	**0.000**	0.952	0.382
Bacteroidetes; Bacteroidia; Bacteroidales; Rikenellaceae	0.0859869	0.098791	0.074076	0.0002203	0.0002487	0.0006518	**0.028**	0.952	0.129
Cyanobacteria; Vampirovibrio; FR888536; FR888536	0.0017606	0.000361	0.00032	0	0	0	**0.000**	0.952	0.437
Firmicutes; Bacilli; Lactobacillales; Streptococcaceae	0.0004335	0.000424	0.000158	0.000057	0	0.0000121	*0.096*	0.952	**0.000**
Firmicutes; Clostridia; Clostridiales;	0.0000972	0.000076	0.000134	0	0	0	**0.017**	0.952	*0.057*
Firmicutes; Clostridia; Clostridiales; Mogibacterium	0.0001579	0.000209	0.000113	0	0	0	**0.007**	0.952	0.382
Firmicutes; Erysipelotrichi; Erysipelotrichales; Erysipelotrichaceae	0.000253	0.000271	0.000353	0	0	0.0001831	**0.000**	0.952	*0.057*
Proteobacteria; Gammaproteobacteria; Enterobacterales; Enterobacteriaceae	0.0002524	0.000115	0.000465	0.000057	0.000388	0.0017591	**0.028**	0.952	0.449
**Day 14**									
Actinobacteria; Coriobacteriia; Coriobacteriales; Coriobacteriaceae	0.000918	0.001082	0.0010444	0.0000151	0.0000356	0.0004526	**0.000**	0.619	0.284
Bacteroidetes; Bacteroidia; Bacteroidales; AC16063Unknown	0.01686	0.013505	0.0166571	0.0000033	0	0.0000054	**0.000**	0.798	0.664
Bacteroidetes; Bacteroidia; Bacteroidales; Muribaculaceae	0.375628	0.498902	0.572867	0.410043	0.133806	0.14136	0.433	**0.000**	0.202
Bacteroidetes; Bacteroidia; Bacteroidales; Rikenellaceae	0.083308	0.095564	0.0898889	0.0015447	0.0000106	0.105484	**0.046**	0.619	0.202
Cyanobacteria; Vampirovibrio; FR888536; FR888536	0.000882	0.00029	0.000229	0.0000033	0	0	**0.021**	0.652	0.277
Firmicutes; Bacilli; Lactobacillales; Lactobacillaceae	0.043107	0.044722	0.125749	0.0000103	0.0000029	0.0487264	**0.004**	0.619	**0.000**
Firmicutes; Bacilli; Lactobacillales; Streptococcaceae	0.000322	0.000367	0.0001771	0	0	0	**0.000**	0.619	0.205
Firmicutes; Clostridia; Clostridiales; Lachnospiraceae	0.284822	0.154822	0.0736786	0.445336	0.325295	0.151596	**0.002**	0.176	**0.016**
Firmicutes; Clostridia; Clostridiales; Mogibacterium	0.000224	0.000167	0.0000435	0.0000022	0	0	**0.000**	0.619	*0.065*
Firmicutes; Clostridia; Clostridiales; Peptococcaceae	0.00015	0.000107	0.0000104	0.0000143	0	0	**0.002**	0.619	**0.016**
Proteobacteria; Gammaproteobacteria; Enterobacterales; Enterobacteriaceae	0.000166	4.28 × 10^−5^	0.0000684	0.0008475	0.196246	0.082846	0.550	**0.008**	0.205
**Day 21**									
Bacteroidetes; Bacteroidia; Bacteroidales; AC16063Unknown	0.0198385	0.0155954	0.0136087	0.0000043	0.0000537	0	**0.000**	0.813	0.632
Bacteroidetes; Bacteroidia; Bacteroidales; Bacteroidaceae	0.0294528	0.0253085	0.0085189	0.0954284	0.165391	0.0475824	**0.044**	0.221	**0.034**
Bacteroidetes; Bacteroidia; Bacteroidales; Rikenellaceae	0.117293	0.078363	0.081423	0.0128481	0.0146536	0.0796541	**0.003**	0.414	0.507
Firmicutes; Bacilli; Lactobacillales; Lactobacillaceae	0.0550488	0.0971074	0.125959	0.0000263	0.0260096	0.0193837	**0.003**	0.813	0.507
Firmicutes; Clostridia; Clostridiales	0.0000553	0.0002943	0.0002156	0.0000234	0.0000111	0.0000502	**0.044**	0.813	0.637
Firmicutes; Clostridia; Clostridiales; Lachnospiraceae	0.185813	0.17024	0.149844	0.337789	0.174787	0.17705	**0.000**	**0.008**	0.750
Firmicutes; Clostridia; Clostridiales; Peptococcaceae	0.0001432	0.0000541	0.0000536	0.0002975	0	0	**0.044**	**0.019**	0.750
Firmicutes; Clostridia; Clostridiales; Ruminococcaceae	0.0565782	0.0509838	0.0472314	0.0920033	0.0277662	0.0477612	**0.008**	**0.000**	0.750
Firmicutes; Erysipelotrichi; Erysipelotrichales; Erysipelotrichaceae	0.0001714	0.0001399	0.0002673	0.0004117	0.0031187	0.0011978	0.751	**0.019**	0.507

## Data Availability

Data are stored at https://osf.io/42gz6/ (accessed on 6 May 2025).

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
