# Peer review of "The Impact of Human Milk Oligosaccharides on Antibiotic-Induced Microbial Dysbiosis and Gut Inflammation in Mice"

_antibiotics, 2025, doi:10.3390/antibiotics14050488_

Round 1
Reviewer 1 Report
Comments and Suggestions for Authors
Review report
Lines 35–43: Delete these sentences, as these are author instructions.
Lines 65–68: Rewrite this sentence. Please simplify the most complex and long sentences.
Lines 72–73: Write complete mannan-oligosaccharides (MOS) and xylose-oligosaccharides (XOS) instead of their short names.
Lines 97–100: Rewrite this sentence. Please simplify the most complex and long sentences.
In the introduction, it must provide the most recent reference, i.e., after 2020 (only 8 references are the latest from 23 references).
Lines 130–133, and 175–186, Please simplify the most complex and long sentences.
Statistical significance markers might be better explained by some figure legends.
Explain the exclusion of certain post hoc tests (such as those for water consumption).
Line 346, Explain the rationale for the use of just female mice and how this might restrict generalizability.
Line 371, It is necessary to justify the ampicillin dosage of 1 g/L in comparison to human counterparts.
Lines 419–420 must cite a peer-reviewed article reference.
Although the results are properly interpreted in the discussion, the constraints may be more effectively integrated.
Line 280, Though intriguing, the conjecture regarding lactoferrin's function needs more concrete proof or references.
The findings support the conclusion, but they can also highlight the necessity for human research.
Comments on the Quality of English LanguageThe English language and quality of writing are moderate.
Some sentences have grammatical errors; for example, verb tenses are incorrect.
Please simplify the most complex and long sentences. In multiple areas, you have used improper punctuation.
It must be completely rewritten in English, ideally by an editing agency or native English speaker.
Author Response
We thank the reviewer for these constructive comments.
Lines 35–43: Delete these sentences, as these are author instructions.
- Done
Lines 65–68: Rewrite this sentence. Please simplify the most complex and long sentences.
- We have rewritten the sentence, so it now reads: ‘In mammals up to 1013 bacteria may have colonized the gut [9], supported by oral tolerance maintained by the gut immune system [10]. It may, therefore, be optimistic to expect a few beneficial bacteria to colonise in such densely populated microbiotic en-vironment. Although engineered probiotics may hold promise as future therapeutics, their application still faces several challenges [11].'
Lines 72–73: Write complete mannan-oligosaccharides (MOS) and xylose-oligosaccharides (XOS) instead of their short names.
- We have removed the abbreviations of the plant oligosaccharides.
Lines 97–100: Rewrite this sentence. Please simplify the most complex and long sentences.
- We have revised the sentence so it now reads: ‘In this study, we treated mice with ampicillin and additionally administered 2′-FL, either alone or in combination with DFL. We hypothesised that these HMOs might alleviate gut microbiota dysbiosis. Our primary readout was microbiota characterization via 16S amplicon sequencing. Secondary readouts included assessment of intestinal cytokines, gene expression, and SCFA concentrations. The mixture of 2’FL and DFL appeared to mitigate some of the adverse effects of ampicillin treatment.’
In the introduction, it must provide the most recent reference, i.e., after 2020 (only 8 references are the latest from 23 references).
- We have updated all references to make sure that it is what we can find as the newest ones.
Lines 130–133, and 175–186, Please simplify the most complex and long sentences.
- Ln 130-133 now reads: ‘Bray-Curtis indexes (beta-diversity) express the differences between the microbiota composition of each individual mouse. The closer the mice cluster the smaller the difference. Ampicillin-treated mice clustered different from non-treated mice on day 7, 14 and 21 (Figure 2B). There was no significant HMO-specific clustering of the ampicillin-treated mice, while the mice not administered ampicillin clustered significantly different according to being HMO-treated or not (Figure 2B).’
- Ln 175-186 now reads: ‘Ampicillin significantly reduced the relative abundances of a large number of bacterial families. The 2’FL+DFL mixture significantly increased the relative abundances of certain bacterial families. The effect was more limited when 2’FL was administered alone (Table 1 and S1), though its influence remained detectable - particularly in mice treated solely with ampicillin- even one week (Figure 3), two weeks (Figure 4), and three weeks (Figure 5) after cessation of treatment. On day 14, i.e. two weeks after the cessation of ampicillin treatment, the ampicillin induced reduction in Lactobacillaceae and Coriobacteriaceae appeared to be counteracted by the 2’FL + DFL mixture (Figure 4 A and B). This effect was at the genus level driven by Lactobacillus spp. (Figure 6). Additionally, Bacteroides spp. were promoted in ampicillin-treated mice receiving the 2’FL + DFL mixture (Figure 6). On day 14, both HMOs appeared to increase the abundance of Enterobacteriaceae in ampicillin-treated mice (Table 1; Figure 4C). This effect was at the genus level driven by Escherichia spp. (Figure 6).’
Statistical significance markers might be better explained by some figure legends.
- We agree that on Figures 7, 8 and 9 it should have been explained that values on the graphs are p-values from Tukey’s posthoc test. This has been done now.
Explain the exclusion of certain post hoc tests (such as those for water consumption).
- Generally, for the statistics post hoc tests are only omitted, if there is not an overall significant p or q value, and that is a standard statistical principle. For the water consumption the hypothesis was that ampicillin and/or one of the two HMO mixtures influenced water intake, and that is answered by the overall p-values in the three-way ANOVA. It is not necessarily good statistical practice to make pot hoc tests after a multi-factorial ANOVA if the hypothesis already has been answered, as it violates the principle of only making statistical tests for specified hypotheses. Furthermore, water intake is monitored on cage level, and, therefore, group sizes in a post hoc Tukey’s test would be n=2, and that does not make sense to compare.
Line 346, Explain the rationale for the use of just female mice and how this might restrict generalizability.
- We have introduced the sentence ‘The rationale for this was to prevent cage fighting, but this unfortunately limits the validity of results for both sexes..
Line 371, It is necessary to justify the ampicillin dosage of 1 g/L in comparison to human counterparts.
- We have inserted the following sentence: ‘This ampicillin dose known to be efficient, as it is used in most studies on microbiota eradication in mice [60]. Compared to a human dose, t is in the high end if recalculated for mice according to metabolic weight [61].’
Lines 419–420 must cite a peer-reviewed article reference.
- We have updated these references.
Although the results are properly interpreted in the discussion, the constraints may be more effectively integrated.
- We understand this input, and we have rewritten the discussion; also to integrate some of the constraints. However, we find it is important that the limitations of a study are clearly presented in its own separate section.
Line 280, Though intriguing, the conjecture regarding lactoferrin's function needs more concrete proof or references.
- We agree that this is speculative, and we have deleted this part.
The findings support the conclusion, but they can also highlight the necessity for human research.
- We agree. We have inserted the following in the conclusion: ‘The potential of HMOs for minimizing the adverse effects of ampicillin should be further studied in human patients.’
Comments on the Quality of English Language
The English language and quality of writing are moderate.
Some sentences have grammatical errors; for example, verb tenses are incorrect.
Please simplify the most complex and long sentences. In multiple areas, you have used improper punctuation.
It must be completely rewritten in English, ideally by an editing agency or native English speaker.
- We have supported by the department translator rewritten the manuscript, and we hope that the English language is now satisfactory. This also includes shortening of sentences.
Reviewer 2 Report
Comments and Suggestions for Authors
This paper can be publish under antibiotics . The author describe about the effect of ampicillin inducing microbial dysbiosis and also the impact of HMO in gut microbiota. author manage to direct reader based on the aim of the study manage to answer with appropriate method design and results. Well explained research paper with appropriate diagram and results. Just some minor correction as below.
1. Kindly remove part from line 35- 43. Should not include in introduction, is meant for author references.
2. Line 251 , the sentences is incomplete, please update it.
3.Line 259, add in comma after "ampicillin treatment, it also"..
4. For conclusion I am aware that the author included it under discussion section, possible separate your conclusion in a new heading.
5.Line for reference no1 , please change to recent publication.
6. Line 53-54, add recent references to this sentences.
7. Please check your references entirely, include a recent publication work for your references.
8. Please include further direction, of your research and what was author will like to look forward in next research based on this manuscript.
9. for keyword, can add HMO
Author Response
We thank the reviewer for the positive attitude to our paper and the constructive comments.
This paper can be publish under antibiotics . The author describe about the effect of ampicillin inducing microbial dysbiosis and also the impact of HMO in gut microbiota. author manage to direct reader based on the aim of the study manage to answer with appropriate method design and results. Well explained research paper with appropriate diagram and results. Just some minor correction as below.
- Kindly remove part from line 35- 43. Should not include in introduction, is meant for author references.
- Done
- Line 251 , the sentences is incomplete, please update it.
- We apologize for this. The sentence now reads: ’ in the early phase immediately after cessation of ampicillin treatment, it also mitigated the ampicillin-induced reduction in gut microbiota diversity..’
3.Line 259, add in comma after "ampicillin treatment, it also"..
- Done
- For conclusion I am aware that the author included it under discussion section, possible separate your conclusion in a new heading.
- Done
5.Line for reference no1 , please change to recent publication.
- We have updated this to a more recent publication (Marione et al, 2021).
- Line 53-54, add recent references to this sentences.
- We have updated with the most recent references.
- Please check your references entirely, include a recent publication work for your references.
- We have updated references with the most recent references, so we now have app. half of the references after 2020, and two thirds of the references after 2015.
- Please include further direction, of your research and what was author will like to look forward in next research based on this manuscript.
- We now end the conclusion as follows: ‘The potential of HMOs for minimizing the adverse effects of ampicillin should be further studied in human patients. It would also be relevant with further mouse studies in which some other HMOs, e.g. some sialylated HMOs, are applied.’
- for keyword, can add HMO
- HMO is not listed as a MESH term
Reviewer 3 Report
Comments and Suggestions for Authors
Dear Authors,
Your work in this study is much appreciated.
I would like to share some questions and points for your consideration.
Best of luck.
General comments
A well-designed and strong methodological framework. This research is significant and current due to the growing focus on prebiotics as alternatives or supplements to probiotics for re-establishing gut microbial equilibrium after antibiotic therapy.
Important questions and comments.
- The authors recognize during the discussion that mouse microbiota presents substantial differences from human microbiota, specifically regarding HMO metabolism. The study should explain this limitation about mouse microbiota differences from human microbiota in the Introduction to properly establish the translational value of their findings.
- HMO doses were determined by the authors through metabolic weight conversion calculations. It remains uncertain if the dosages studied reflect relevant physiological levels seen in human infants. Does preliminary dose-finding research verify the selected concentrations?
- Bifidobacterium spp. Researchers identify Bifidobacterium spp. solely at the genus level, even though they play a critical role in HMO metabolism. Without strain-level identification of HMO utilization by Bifidobacterium, it becomes difficult to draw firm conclusions about the absence of their response.
- Some phrasing is overly informal for a scientific manuscript. For instance: “We gave six groups of 8 mice drinking water…” → Consider rephrasing to: “Six groups of eight mice were administered…”
- Please provide further explanation or data to reinforce the hypothesis that increased fecal Escherichia spp. does not exhibit pathogenic characteristics in this situation. is not pathogenic in this context?
- The study chose 2′-FL and DFL instead of other HMOs because? Were others considered (e.g., sialylated HMOs)?
- The researchers conducted preliminary tests to identify the best HMO concentration for anti-inflammatory results.
- The authors should explain why 2′FL alone produced minimal results when compared to the combination of 2′FL+DFL.
General revision needed.
Author Response
We thank the reviewer for the positive and constructive review.
General comments
A well-designed and strong methodological framework. This research is significant and current due to the growing focus on prebiotics as alternatives or supplements to probiotics for re-establishing gut microbial equilibrium after antibiotic therapy.
Important questions and comments.
1. The authors recognize during the discussion that mouse microbiota presents substantial differences from human microbiota, specifically regarding HMO metabolism. The study should explain this limitation about mouse microbiota differences from human microbiota in the Introduction to properly establish the translational value of their findings.
- We have inserted the following text in the introduction: ‘Mice are an easy and inexpensive tool for preclinical studies under well-controlled conditions prior to more complex and costly trials in human patients, although within certain research areas their validity may be low [28]. In addition, to the advantage of having enough units to allow for proper statistical analysis, another benefit of using mice is the existence of a vast body of literature demonstrating the efficacy of oligo-saccharides in mice [24,29-34]. However, a major disadvantage is that mouse milk pri-marily contains sialylated oligosaccharides, whereas human milk primarily contains fucosylated oligosaccharides, which specifically target different bacteria [32].’
2. HMO doses were determined by the authors through metabolic weight conversion calculations. It remains uncertain if the dosages studied reflect relevant physiological levels seen in human infants. Does preliminary dose-finding research verify the selected concentrations?
- What we can say is that there are good data on the HMO content in human milk and on the intake in human infants. It is normal practice to recalculate dosing according to metabolic weight in mouse studies, but there is very little literature information, whether this is actually the most efficient dose in mice. We have rewritten to better explain the background for a human dose: ‘). HMO dose was calculated from the assumption that each mouse drinks approximately 3.5 ml water per day and weighs approximately 20 g during the study. Mature human milk contains 10-15 gram HMO per liter [65] and the mean daily intake for an infant is 670 ml [66], which gives an average intake of approximately 8-10 g per day. Recalculation of a mouse dose from a human dose of 10 g per day according to metabolic weight [64] gives a dose of 140 mg/kg body weight. Our HMO dose is equivalent to anti-inflammatory doses in other studies [33]. However, is not uncommon in mouse studies to use doses much higher than what is human equivalent [34,67].’
3. Bifidobacterium spp. Researchers identify Bifidobacterium spp. solely at the genus level, even though they play a critical role in HMO metabolism. Without strain-level identification of HMO utilization by Bifidobacterium, it becomes difficult to draw firm conclusions about the absence of their response.
- We agree with the reviewer, but we have already mentioned this as a limitation. We use sequencing with rather high resolution, but this does not necessarily implicate that certain OTUs can be identified to a species level in mice at the same rate as in humans, as the information in the databases is still far more detailed for humans than for mice.
4. Some phrasing is overly informal for a scientific manuscript. For instance: “We gave six groups of 8 mice drinking water…” → Consider rephrasing to: “Six groups of eight mice were administered…”
- We have rephrased as proposed, and we have generally given the manuscript a linguistic make-over.
5. Please provide further explanation or data to reinforce the hypothesis that increased fecal Escherichia spp. does not exhibit pathogenic characteristics in this situation. is not pathogenic in this context?
- We observe that increased abundance of Eschericia does not increase inflammation. Due to what is discussed next the importance of a high abundance of Eschericia should not be over-interpreted, as we know that oligosaccharides bind Enterobacteriaceae, and, therefore, a high abundance in the feces of HMO treated mice might be expected, but it may not necessarily reflect a propagation.
6. The study chose 2′-FL and DFL instead of other HMOs because? Were others considered (e.g., sialylated HMOs)?
- There are good chances that sialylated HMOs could have more dramatic effects in mice, but they are not so human relevant, as human milk is dominated by fucosylated HMOs. But it is relevant to study in future studies and this has been mentioned under 'Conclusions'.
7. The researchers conducted preliminary tests to identify the best HMO concentration for anti-inflammatory results.
- Doses within this range have previously shown to have an anti-inflammatory impact on mice; please se our reply to point 2.
8. The authors should explain why 2′FL alone produced minimal results when compared to the combination of 2′FL+DFL.
- It is difficult to explain from this study, but there is a similar study on visceral hypersensitivity and dysbiosis, in which the combination of HMO’s have been shown to be the most efficient. We have inserted a sentence in the discussion Ln 279-282: ‘It has previously been shown that the combination of two HMO’s is the most efficient for eliciting effects through the gut microbiota [41], but it is difficult from this and our study to explain the mechanism behind this.’
Comments on the Quality of English Language
General revision needed.
- As also noted to reviewer 1, we have supported by the department translator rewritten the manuscript, and we hope that the English language is now satisfactory. This also includes shortening of sentences.
Round 2
Reviewer 1 Report
Comments and Suggestions for Authors
The work is well-written and has a solid scientific foundation. Please improve readability and more openly address experimental limits.
Line 64: replace word “chemical” with “chemically defined”
Line 98: Replace the word “which specifically target” with “which preferentially support”. Also replace word “bacteria” with “bacterial taxa”
Line 134: Replace the word “different” with “differently”.
Line 250: Remove the word “on” after effect. So, the line should be “effect in all”.
Lines 314 to 317: This statement is a little wordy and unclear. Replace whole statement with this statement: “By mimicking host sugar receptors [19], HMOs probably encourage the accumulation of Enterobacteriaceae in the feces during the early stages of treatment.”
Line 335: Provide a reference for this rationale.
Line 375: Replace “t” with “it”.
Comments on the Quality of English LanguageIt needs slight English language rephrasing.
Author Response
We thank the reviewer for once again taking the time to deliver constructive input to our paper.
We have the following comments:
The work is well-written and has a solid scientific foundation.
- Thank you.
Please improve readability and more openly address experimental limits.
- We think we have already made the limitations of our study very clear (ln 329 - 339). However, to underline that these limitations of course may influence the predictive validity of the study, we added this sentence (Ln 335-337): 'Overall, this limits the predictive validity of the study, as it remains unclear whether the bacterial taxa affected in mice would also be similarly affected in humans.' Furthermore, in the 'Conclusion' section we have added in as follows (Ln 346-347): 'The potential of HMOs to mitigate the adverse effects of ampicillin should be further investigated in human patients to determine whether the efficacy observed in mice is translatable to humans.'
Line 64: replace word “chemical” with “chemically defined”
- Done
Line 98: Replace the word “which specifically target” with “which preferentially support”. Also replace word “bacteria” with “bacterial taxa”
- Done
Line 134: Replace the word “different” with “differently”.
- Done
Line 250: Remove the word “on” after effect. So, the line should be “effect in all”.
- Done
Lines 314 to 317: This statement is a little wordy and unclear. Replace whole statement with this statement: “By mimicking host sugar receptors [19], HMOs probably encourage the accumulation of Enterobacteriaceae in the feces during the early stages of treatment.”
- Done
Line 335: Provide a reference for this rationale.
-Done
Line 375: Replace “t” with “it”.
- Done
It needs slight English language rephrasing.
- We admit that especially the results section could have been better English, and we have tried the best of our abilities to improve that. We have also made some minor linguistic correction in the revised manuscript.